# An Investigation of Travel Distance and Timeliness of Breast Cancer Treatment Among a Diverse Cohort in the United States

**DOI:** 10.3390/ijerph22020176

**Published:** 2025-01-27

**Authors:** Swann Arp Adams, Oluwole Adeyemi Babatunde, Whitney E. Zahnd, Peiyin Hung, Karen E. Wickersham, Nathaniel Bell, Jan M. Eberth

**Affiliations:** 1Department of Biobehavioral Health and Nursing Science, College of Nursing, University of South Carolina, Columbia, SC 29208, USA; kwickers@mailbox.sc.edu; 2Department of Epidemiology & Biostatistics, Arnold School of Public Health, University of South Carolina, Columbia, SC 29208, USA; 3Department of Psychiatry, Prisma Health, Greer, SC 29650, USA; oluwole.babatunde@prismahealth.org; 4Department of Health Policy and Management, College of Public Health, University of Iowa, Iowa City, IA 52242, USA; whitney-zahnd@uiowa.edu; 5Health Services, Policy, and Management Department, Arnold School of Public Health, University of South Carolina, Columbia, SC 29208, USA; hungp@mailbox.sc.edu; 6Rural and Minority Health Research Center, Arnold School of Public Health, University of South Carolina, Columbia, SC 29208, USA; 7Institute for Families in Society, College of Social Work, University of South Carolina, Columbia, SC 29208, USA; nateb@mailbox.sc.edu; 8Department of Health Management and Policy, Drexel University, Philadelphia, PA 19104, USA; jme336@drexel.edu

**Keywords:** breast neoplasm, medical geography, health care disparities, time-to-treatment, African Americans, rural populations

## Abstract

Travel to and from distant cancer treatment facilities can place a significant burden on cancer patients, particularly rural and minority survivors. Thus, the purpose of our investigation was to examine the association between patient travel distance and delays in types of treatment for breast cancer (surgery, radiation, chemotherapy, and hormonal therapy) and treatment delays. Using a novel linkage between the state cancer registry and administrative data from Medicaid and a private insurance plan, 2155 BC patients were successfully linked to create the cohort. ArcGIS was used to geocode all case residences and treatment facility addresses and calculate network distance between the residence and each facility. Logistic regression models were used to calculate the adjusted odds of being delayed versus timely by street distance. Odds of late surgery were increased by 1% (95% CI: 1.00, 1.01) for each one-mile increase from the patient’s residence to the treatment facility. In race-stratified models, the odds of late treatment for Black patients increased by 3% per mile (95% CI 1.01, 1.06) for radiation. Increased travel distance appears to significantly increase treatment delays for surgical, radiation, and chemotherapeutic treatments for women with BC, especially among Black women.

## 1. Introduction

Cross-disciplinary approaches combining geography and medical fields have tremendous potential to bridge significant gaps in breast cancer (BC) research and allow for more comprehensive examination of access to health care. SC has significant racial and geographic disparities for BC, with Black women having significantly lower incidence, but higher mortality [1,2,3]. It has been estimated that Black women experience a 65% greater mortality burden after accounting for incidence compared to White women [2,3].

Increasing evidence is accruing that where patients live and seek medical care can have a profound impact on health outcomes. Celaya et al. showed that women who lived an increasing distance from residence to radiation treatment facility were less likely to have breast-conserving surgery [4]. In addition, a substantial proportion of early-stage BC patients received suboptimal treatment (as defined by National Comprehensive Cancer Network Guidelines) by forgoing radiation because of difficulty traveling to radiation centers [4]. In another study, BC patients who lived at greater distances from a radiation oncology facility were more likely to undergo mastectomy while breast-conservative surgery was negatively correlated with travel distance [5]. Although some studies have examined travel distance and cancer outcomes [6,7], there are limited empirical data available concerning the impact of racial and geographic factors on this association.

Much of the previous literature examining distance and BC treatment has been further limited by the structure of our national cancer registry program [8,9,10]. The SEER-Medicare database is the only national data source available publicly for treatment analyses, and yet SEER only covers approximately half of the US population (and features lower representation of minority and rural populations), and Medicare is limited to only those individuals who are 65 and older. Thus, there is a significant gap in the literature examining travel distance for younger, rural, and minority BC survivors [11]. Furthermore, it is this same population which suffers disproportionately from more aggressive BC disease and mortality.

Race and geographic disparities in survival and adherence to adjuvant hormone treatment (AHT) have been documented in our past research [2,12,13,14]. The average medication possession ratio for AHT was significantly higher among White compared with Black women who had BC [14]. Additionally, Black women who lived in rural areas showed significantly lower adherence to use of AHT [14]. We also showed that there were racial and regional differences in the mortality experienced by BC patients in SC [12]. Consideration of the relationship between travel distance and timeliness of BC treatment with additional assessment of the role of race and geographic determinants is therefore warranted in SC.

Consequently, the purpose of this investigation was to examine the association of travel distance with timeliness of BC treatment. Our primary research question was, “what is the impact of travel distance on timeliness of BC treatment?” We hypothesized that, as travel distance increased, the odds of delayed treatment also increased. Given the large rural and racial BC disparities in our state, we further hypothesized that this effect would be worse among racial and rural minorities.

## 2. Materials and Methods

Applications were completed for each data source (see Figure 1) and reviewed by each corresponding agency specific for the source (cancer registry, Medicaid, and the private insurance plan). Approval was granted for the data use with the provision that only de-identified data be released by the third party trusted data broker [The Revenue and Fiscal Affairs Office (RFA)]. All data received by the investigative team were de-identified and pre-existing; therefore, the study was deemed exempt from IRB review. We conducted a retrospective cohort study using linked data from the SC Central Cancer Registry (SCCCR) with administrative data from Medicaid and a state-based private payor insurer (Figure 1). All female BC patients were initially selected from the cancer registry database using eligibility criteria which included a diagnosis of a first primary, histopathologically confirmed BC, race of either White or Black, and diagnosed between 2002 and 2010 (surveillance period was 2000–2013 for these cases).

The datafile was transmitted to the state integrated data system housed within RFA Data Integration and Analysis Division. Using well-validated probabilistic matching procedures [12,14,15], the BC cases were linked to administrative insurance data. Only those BC records which were linked definitively with either Medicaid or the private payor administrative database were retained. After this initial linkage, additional inclusion criteria were applied such that we only included records for analysis that had a minimum of 3 years of continuous coverage from diagnosis and no dual insurance enrollment. This ensured that a complete record of the patient’s treatment history was captured, and bias was reduced from misclassification of treatment.

The residence at diagnosis for all BC cases was geocoded by the SCCCR to the exact longitude and latitude according to standard case ascertainment protocols. This process involved using the latest address locator and ensuring high accuracy by verifying against multiple data sources. Upon completion of the matching process, RFA geocoded all addresses for service providers listed on the medical and pharmacy files for all linked cases using ArcGIS Pro software, following similar verification steps. With the coordinates of patients and providers, exact network travel distances (i.e., the distance traveled using road transportation, in miles) were then calculated using the Network Analyst extension in ArcGIS Pro.

To calculate the time to treatments while maintaining patient confidentiality (date of service is protected health information), exact age (in days) at each service date was calculated by RFAO. The patients’ age at diagnosis (in days) was also provided by the SCCCR. Hence, the number of days between diagnosis and treatment were calculated by subtracting the age at diagnosis from age at service. Treatment was operationalized using combinations of ICD-9, HPCT (Healthcare Common procedure Coding System), CPT (Current Procedural Terminology), and NDC (National Drug Code) codes for surgery, first chemotherapy, first radiation therapy, and first hormone therapy using algorithms previously validated from the literature [16,17,18,19]. For each treatment type, all records were sorted in ascending order by age at service and the first record appearing after diagnosis for each treatment type was used for the treatment time calculations noted previously. Those who did not receive a specific type of treatment were not included in analyses for that treatment outcome; however, they were included in analyses of other treatment outcomes. The data file that was ultimately provided to the research team was stripped of most identifying variables (with the exception of county) and a unique identifier was assigned to each individual. This allowed for clarification of any data problems or issues with RFA. The key to this unique identifier was only available to RFA.

In initial examinations, the distribution of treatment times for each modality was extremely skewed. Consequently, we classified each treatment time into ‘timely’ or ‘delayed’ according to nationally recognized benchmarks for time from initial diagnosis to standard treatments as recommended by National Comprehensive Cancer Center guidelines (https://www.nccn.org/professionals/physician_gls/default.aspx, accessed on 24 January 2025). This included ≤60 days for surgery [20,21], ≤120 days for chemotherapy [22], ≤365 days for radiation therapy [23], and ≤180 days for hormonal therapy [24].

Additional variables which were considered as confounders and effect modifiers included race (Black/White), rurality (urban/rural), married (yes/no), insurance type (1 vs. 2; encrypted due to payor source requirements for data use), age, stage (advanced/early), and participation in the National Breast and Cervical Cancer Early Detection Program (NBCCEDP; yes/no). Rurality was determined using the 2003 Rural Urban Continuum Codes with 1–4 designated as urban and 5–9 designated as rural. This rural-urban measure categorizes counties based upon their population size and proximity to metropolitan areas and is slightly altered from tradition designations (1–3 vs. 4–9). As this investigation focused on travel distances, we contended that counties adjacent to metropolitan areas were more similar to metropolitan areas than to rural counties without access.

All analyses were conducted using SAS version 9.4. Descriptive statistics including frequencies and means were calculated for all variables. Unconditional logistic regression models [25] were used to compute the odds ratio to assess the relationship between network travel distance and timely/delayed treatment for each modality (surgery, chemotherapy, radiation therapy, and hormone therapy). A ‘full’ model was run with all variables included in the model statement (travel distance, race, rurality, marriage, insurance type, stage, and NBCCEDP participation). As we had an a priori hypotheses for racial and rural disparities, we statistically assessed for this interaction with distance [travel distance + (race or rural variable) + distance*(race or rural variable). We observed evidence for effect modification (interaction term *p*-value ≤ 0.10) by race only; thus, the results for models stratified by race are also presented. Of note, insurance type (encrypted) was included in all models; however, data use agreement contracts prohibited the presentation of these results in our tables. This variable was only included to account for confounding by this variable, hence the measure of association was not relevant to our results and discussion.

## 3. Results

### 3.1. Cohort Description

Table 1 depicts the demographic, tumor, and treatment characteristics of the cohort. Nearly 30% of the cohort was Black and 16% resided in rural counties. Most of the cohort had moderately- or poorly-differentiated breast tumors.

### 3.2. Time to Treatment

The average time from diagnosis to treatment was 56 days for surgery, 152 days for radiation therapy, 88 days for chemotherapy, and 190 days for hormone therapy (Table 2). The average survival time was 82 months. Treatment was delayed for approximately 18%, 2%, 10%, and 49% of the sample for surgery, radiation therapy, chemotherapy, and hormone therapy, respectively.

### 3.3. Regression Model Associations

Logistic model results are shown in Table 3. Statistically significant effect modification was found between race and travel distance in the models for radiation and chemotherapy treatment, but not for surgery or hormonal therapy models. There was no statistically significant effect modification between rural and travel distance for any of the treatment modalities. In overall models, statistically significant associations were found between travel distance for surgery and radiation therapy. For surgery, the odds of receiving delayed surgery increased by 1% for each 1-mile increase in distance traveled. Interestingly, rural and Black patients also had significantly increased odds ratios for delayed surgery, with 1.57 (95% CI = 1.00–2.47) and 1.42 (95% CI = 1.04–1.97), respectively, in this same model.

Race was a significant effect modifier of the relationship between travel distance and late radiation therapy as well as late chemotherapy treatment (*p*-value for interaction 0.10, and 0.08), respectively. After adjustment for rurality, insurance type, marital status, NBCCEDP participation, stage, and age (Table 3), Black women had a 3% increase in odds of late radiation therapy for every 1-mile increase in distance traveled (OR = 1.03; 95% CI = 1.00–1.06). Interestingly, upon stratification by race, no statistically significant association was observed between the distance traveled for care and late chemotherapy. Although the interaction term (race*distance) for late surgery models was not statistically significant, stratified models revealed a statistically significant association between the distance traveled for care and late surgery for Whites, which was not evident among Black BC survivors.

## 4. Discussion

Our findings indicate that increased travel distance may be a barrier to timely surgery and radiation therapy treatments for BC survivors. Because South Carolina is a largely rural state [26,27,28], and given the effect modification of the association by race, these results could point the way for promising interventions to improve the large racial disparities seen for BC mortality in our state [2,12,29,30]. Mortality among White BC patients is 7% lower in SC compared to the national average; however, mortality is 29% higher among Black breast cancer patients in SC [29,30].

We originally hypothesized that surgical treatments would be delayed by increased travel distance; therefore, we were not surprised by the association between increased mileage, rurality, and race with delayed surgical treatment for BC. Previous studies identified no association between rurality and time to surgery for BC, but these studies were situated in Australia with a single-payer health care system [31]. Additionally, researchers have shown that patients who travel further to receive breast conserving/reconstruction surgery compared to radical mastectomy [32], and similarly, women who live closer to their treating facility, are more likely to receive breast-conserving surgery [33]. While our study did not explore the type of BC surgery women received, the association between travel distance and delayed surgical treatment that we identified may be due to the need for patients to be referred to more distant facilities to receive their preferred surgical treatment. Our findings of greater odds of delayed surgical treatment for BC for Black women corroborate previous studies [34,35]. Reducing delays in surgery among rural and Black women is critical, especially as delayed treatment initiation has been associated with decreased survival and poorer outcomes [36].

Transportation barriers are an important aspect of access to cancer care and have been shown to impact outcomes across the cancer control continuum, including late stage at diagnosis, time to treatment initiation, and receipt of or adherence to treatment [37,38,39,40,41,42]. Those living in rural areas are less likely to have access to cancer specialists and higher ranked/designated cancer hospitals [43,44]. Reflecting the compounding impacts of rurality and minority status, studies have also revealed a disproportionate impact of distance to care among Black patients [45,46]. Our findings align with this literature, showing that the odds of late treatment increase as distance to surgery, radiation therapy, and chemotherapy increase. However, the impact of distance hinges on patient race, with Black women experiencing higher odds of late treatment with longer distances to care. Factors such as socioeconomic status and insurance coverage and type may explain some, but not all, of this disparity [13,47]. Structural and health-system changes may be required to overcome these disparities and ensure all women, particularly Black women, receive support and travel assistance to access timely and high-quality care.

Unlike some of our other investigations examining distance and cancer screening [48], rurality was not a significant effect modifier of the relationship between travel distance and any of the treatment outcomes. Additionally, rurality was only significantly associated with treatment delays for surgery. This is not unexpected, as the Commission on Cancer hospitals typically performing these surgeries are sparsely scattered throughout SC (https://www.facs.org/search/cancer-programs?state=SC, accessed on 24 January 2025). We expected to see similar delays for radiation therapy given that facility locations for this type of treatment are similarly sparce in SC. It is unknown if this finding is a result of providers not recommending radiation therapy for rural patients or if there are other underlying causes. To examine this further, future investigations could examine the receipt of stage-specific standardized therapies by rurality. Yet another underlying contributing factor may be the misclassification of urban/rural status for residences. This investigation utilized county-level estimates of rurality which assumes a level of homogeneity across the county which may not be accurate. A smaller area estimation of urban/rural status would be useful to examine this further.

The timeliness of both chemotherapy and hormonal therapy were not impacted by travel distance. This was not unexpected for hormonal therapy, as these medications are available at most commercial pharmacies, even in rural areas. We had hypothesized that chemotherapy timeliness would be influenced by travel distance, so we were surprised by our finding of no association. This may reflect a growing trend of telemedicine, where oncology providers in metropolitan locations prescribe chemotherapy treatment regimens which are administered in local satellite infusion centers or the local primary medical care home of the patient [49]. Other countries, such as Australia and India, with large rural populations have documented similar improvements in the timeliness of cancer treatment [50,51]. More research is needed to understand the underlying mechanisms.

The persistent travel burdens for South Carolina BC survivors to treatments might exacerbate the existing rural and racial disparities in cancer outcomes [1]. Previously, we have documented large disparities in BC staging, adherence to treatment, and survival for both Black and rural BC women [2,3,12,29,52]. Survival might be worsened due to delayed treatment initiation [53], but also be improved when patients optimize their choice of providers and gravitate toward distant tertiary centers [54]. Our findings that Black women had increased delayed treatments with farther travel distance to their cancer facilities suggested that travel distances would not discount White women’s timeliness of cancer treatments. Such racial disparities in the amenability of travel distances suggest a need to develop community-wide initiatives to overcome access barriers associated with travel burdens. For example, the American Cancer Society’s Road To Recovery program (https://www.cancer.org/treatment/support-programs-and-services/patient-transportation.html) and South Carolina Medicaid beneficiaries’ South Carolina Non-Emergency Medical Transportation policy both helped coordinate transportation for eligible patients (https://www.cancer.org/support-programs-and-services/road-to-recovery.html, accessed on 24 January 2025). In addition to these transportation accommodations, health professionals who diagnose cancers also play an essential role in avoiding delayed cancer treatments. Well-planned cancer care at the office visits by considering how far each patient is required to travel given her prognosis would facilitate timely access to appropriate care.

As with any investigation, our results should be interpreted within the context of its limitations. As is usual in much of the GIS literature, individuals listing residential addresses as a post office box are excluded from the models because PO Boxes do not represent physical locations where people live and work. Thus, we were not able to include these patients in our analysis; however, our SC cancer registry maintains gold level certification standards requiring complete case ascertainment (including residential address) for greater than 97.5% of the total sample, which did have valid address information associated with the patient record. Consequently, we do not expect this to bias our study findings. Another limitation is our inability to examine cultural beliefs and myths about cancer that may account for decisions to seek treatment. This would be a ripe area for further research.

While some may see our inclusion criteria of 3 years of continuous insurance enrollment post-BC diagnosis as a limitation, we view this as a strength of our results, as this will minimize bias from treatment misclassification for our study. Another strength which makes a significant contribution to the distance literature is the fact that we were able to calculate exact travel distances between a given service provider and their patient’s residence. The majority of travel distance and care literature calculates distance to the nearest provider (without knowing which provider the patient actually saw) or distance to the zip code centroid of the provider. Hence, our findings are more likely to reflect the actual lived experience of the patient. Other strengths of our investigation include the racially, geographically, and economically diverse sample, as well as the inclusion of a younger sample of BC patients who experience the larger burden of disparities in SC [2]. The only public-use dataset which combines administrative and cancer registry is the SEER-Medicare data. Only a few states have policies and infrastructure which will support linkages between state cancer registries and administrative data; thus, this is one of the few investigations which can focus on women less than 65 years of age.

## 5. Conclusions

Our work highlights a promising area in cancer treatment which could significantly improve racial disparities in SC through interventions which would minimize the impact of travel distance on treatment delays. An important next step for future analysis would be to examine the impact of travel distance on combinations of therapy. Additional research is also recommended to supplement these findings utilizing qualitative methods which could further explore this phenomenon from a patient and provider perspective. Finally, another useful line of future inquiry would be to explore this phenomenon in other rural populations in other geographic regions of the US to confirm generalizability beyond the rural Southern US.

Ideally, interventions developed from this body of work should be aimed at multiple levels, e.g., individual, communities, and policy, to enhance impact. Policy-level interventions have shown to be critical to expanding access to care in rural populations. Examples of this include expanding the scope of practice for pharmacists to allow for the administration of vaccines [55], including HPV vaccination for cervical cancer prevention, and the creation of community oncology research programs by the National Cancer Institute (https://www.ncorp.cancer.gov/about/, accessed on 24 January 2025), which formalizes partnerships between comprehensive cancer centers and community oncology/satellite centers. Additional work is needed to begin to understand how these levels interact with each other and how interventions can best address the needs of BC survivors.

## Figures and Tables

**Figure 1 ijerph-22-00176-f001:**
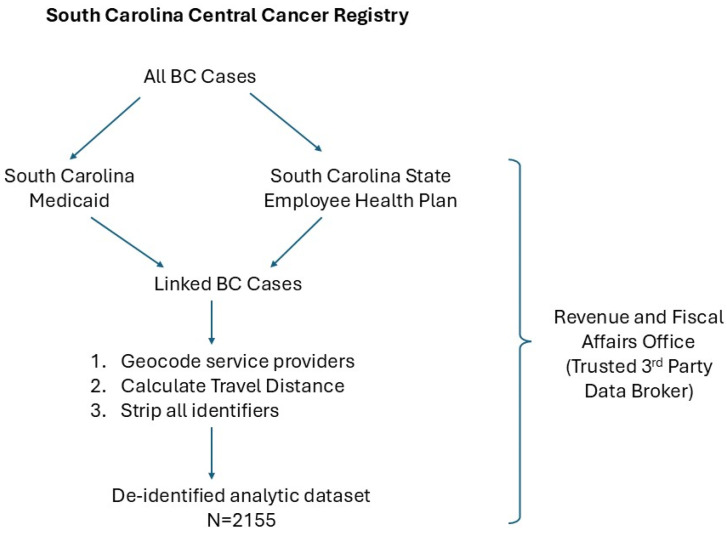
Protocol for creation of final analytic dataset.

**Table 1 ijerph-22-00176-t001:** Breast cancer survivor cohort characteristics, 2000–2013.

Characteristics	% (n) (N = 2155)
Race	
Black	27.8 (598)
White	72.2 (1557)
Rurality	
Rural	15.8 (341)
Urban	84.2 (1814)
Married	
Yes	61.5 (1256)
No	38.5 (787)
NBCCEDP Participant	
Yes	5.5 (119)
No	94.5 (2036)
Grade	
Well Differentiated	18.2 (392)
Moderately Differentiated	36.4 (785)
Poorly Differentiated	34.8 (749)
Undifferentiated	1.5 (32)
Unknown	9.1 (197)
Stage	
0, I, II	88.8 (1767)
III, IV	11.2 (1989)
ER Status	
Positive/Borderline	75.6 (899)
Negative	24.4 (291)
PR Status	
Positive/Borderline	64.8 (768)
Negative	35.2 (417)
Insurance Type	
Type 1	76.1 (1640)
Type 2	23.9 (515)
Received Surgery	
Yes	91.8 (1979)
No	8.2 (176)
Received Radiation Therapy	
Yes	67.0 (1443)
No	33.0 (712)
Received Chemotherapy	
Yes	57.5 (1239)
No	42.5 (916)
Received Hormone Therapy	
Yes	71.1 (1532)
No	28.9 (623)
Vital Status	
Alive	92.2 (1986)
Deceased	7.8 (169)

NBCCEDP = National Breast and Cervical Cancer Early Detection Program; ER = estrogen receptor, PR = progesterone receptor; HT = hormonal therapy.

**Table 2 ijerph-22-00176-t002:** Survival and treatment characteristics of the BC cohort, 2000–2013.

Characteristics		Mean ± SD	
Age at diagnosis (years)		51.2 ± 7.2	
**Treatment Type**	**Receiving Treatment** **% (n)**	**Time from Diagnosis to First Treatment (days)**	**Delayed Treatment** **% (n)**
Surgery	91.8 (1979)	56 ± 113	17.8 (343)
Radiation Therapy	67.0 (1443)	152 ± 123	2.4 (35)
Chemotherapy	57.5 (1239)	88 ± 132	10.3 (127)
Hormone Therapy	71.1 (1532)	190 ± 175	49.3 (742)
		**Time from Diagnosis to Death or Censoring (months)**	
Survival		82.3 ± 31.6	

**Table 3 ijerph-22-00176-t003:** Full models * (overall and race-stratified) of the association between network distance and odds of late/early treatment for surgery and radiation therapy, 2000–2013.

Treatment Outcome	Variable	OVERALL Odds Ratio (95% CI) of Late Treatment	BLACK Odds Ratio (95% CI) of Late Treatment	WHITE Odds Ratio (95% CI) of Late Treatment	*p*-Value for Race * Distance Interaction	*p*-Value for Rural * Distance Interaction
Surgery						
	Travel Distance **	**1.01 (1.00, 1.01)** †	1.00 (0.99, 1.01)	**1.01 (1.00, 1.02)**	0.23	0.14
	Rurality					
	Rural	**1.57 (1.00, 2.47)**	1.08 (0.55, 2.11)	**1.98 (1.04, 3.76)**		
	Urban	1.00	1.00	1.00		
	Race					
	Black	**1.42 (1.04, 1.97)**	--	--		
	White	1.00	--	--		
	Married					
	Yes	1.03 (0.75, 1.41)	1.23 (0.72, 2.12)	0.94 (0.63, 1.38)		
	No	1.00	1.00	1.00		
	NBCCEDP Participation					
	Yes	0.99 (0.47, 1.97)	0.50 (0.20, 1.27)	2.15 (0.68, 6.78)		
	No	1.00	1.00	1.00		
	Stage					
	High	**0.33 (0.23, 0.47)**	**0.36 (0.19, 0.70)**	**0.31 (0.20, 0.48)**		
	Low	1.00	1.00	1.00		
	Age	**0.97 (0.95, 0.99)**	0.98 (0.95, 1.01)	**0.96 (0.94, 0.99)**		
Radiation Therapy						
	Travel Distance **	**1.02 (1.00, 1.03)**	**1.03 (1.00, 1.06)**	1.00 (0.96, 1.03)	0.10	0.49
	Rurality					
	Rural	0.76 (0.26, 2.21)	0.55 (0.12, 2.46)	1.49 (0.18, 12.15)		
	Urban	1.00	1.00	1.00		
	Race					
	Black	1.36 (0.56, 3.32)	--	--		
	White	1.00	--	--		
	Married					
	Yes	0.54 (0.21, 1.41)	0.52 (0.09, 2.96)	0.66 (0.20, 2.12)		
	No	1.00	1.00	1.00		
	NBCCEDP Participation ^ꝉ^	--	--	--		
	Yes	--	--	--		
	No					
	Stage					
	High	**0.36 (0.14, 0.87)**	**0.21 (0.05, 0.87)**	**0.64 (0.17, 2.42)**		
	Low	1.00	1.00	1.00		
	Age	0.97 (0.92, 1.02)	0.94 (0.87, 1.01)	1.01 (0.93. 1.10)		
Chemotherapy						
	Travel Distance **	1.00 (0.99, 1.01)	1.01 (0.99, 1.03)	1.00 (0.99, 1.01)	0.08	0.60
	Rurality					
	Rural	0.75 (0.42, 1.35)	0.45 (0.19, 1.07)	1.15 (0.48, 2.76)		
	Urban	1.00	1.00	1.00		
	Race					
	Black	1.09 (0.68, 1.74)	--	--		
	White	1.00	--	--		
	Married					
	Yes	0.77 (0.48, 1.22)	0.96 (0.42, 2.23)	0.71 (0.40, 1.25)		
	No	1.00	1.00	1.00		
	NBCCEDP Participation					
	Yes	1.72 (0.71, 4.19)	3.91 (0.82, 18.67)	0.85 (0.27, 2.67)		
	No	1.00	1.00	1.00		
	Stage					
	High	1.58 (0.87, 2.88)	1.18 (0.41, 3.34)	1.62 (0.77, 3.40)		
	Low	1.00	1.00	1.00		
	Age	1.03 (1.00, 1.06)	1.02 (0.97, 1.06)	1.03 (0.99, 1.08)		
Hormone Therapy						
	Travel Distance **	1.00 (0.99, 1.01)	1.00 (0.98, 1.02)	1.00 (0.99, 1.01)	0.51	0.67
	Rurality					
	Rural	0.76 (0.54, 1.08)	0.68 (0.35, 1.31)	0.82 (0.54, 1.26)		
	Urban	1.00	1.00	1.00		
	Race					
	Black	1.31 (0.98, 1.75)	--	--		
	White	1.00	--	--		
	Married					
	Yes	1.06 (0.80, 1.39)	0.96 (0.57, 1.62)	1.10 (0.80, 1.53)		
	No	1.00	1.00	1.00		
	NBCCEDP Participation					
	Yes	0.80 (0.47, 1.38)	0.84 (0.37, 1.90)	0.75 (0.36, 1.56)		
	No	1.00	1.00	1.00		
	Stage					
	High	**0.21 (0.13, 0.33)**	**0.34 (0.14 0.85)**	**0.18 (0.11, 0.31)**		
	Low	1.00	1.00	1.00		
	Age	**0.95 (0.93, 0.97)**	**0.94 (0.91, 0.98)**	**0.95 (0.93, 0.97)**		

* Due to the provision of our data agreement contracts, insurance type was included in all analyses, but results cannot be displayed. ** Units are interpreted for a 1-mile increase in distance. ꝉ This variable caused model separation due to small sample size and could not be included in radiation therapy models. † Bolding of estimates indicates statistical significance ≤ 0.05.

## Data Availability

The data used for analysis is not owned by the research team and is governed by a data use agreement which prohibits the sharing of any data files. All programs used for analysis are available upon request from the corresponding author.

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
