# Peer review of "An Investigation of Travel Distance and Timeliness of Breast Cancer Treatment Among a Diverse Cohort in the United States"

_ijerph, 2025, doi:10.3390/ijerph22020176_

Round 1
Reviewer 1 Report
Comments and Suggestions for Authors
I greatly respect the SC Center and the Senior Researchers. Unfortunately, I think the article can be greatly improved by adding a Geospatial Expert. There are many nuances to seeking cancer treatment, including the kind of treatment characteristics that may have been surgery + radiation, Surgery + chemotherapy, Surgery + hormonal therapy, or just surgery alone, because the margins were "clean". What about "White + Non-Hispanic" vs. "White + Hispanic"?
On Page 2, line 52, the authors use the term, "suboptimal treatment". How is this determined, and based on what? Working for a NCI Cancer treatment center, this could be a pretty high standard, or not. Cite your treatment standards.
Working for a NCI Cancer Treatment Center, I am also aware that patients will come, and stay for long periods of time for treatment.
Treatment distances is a major issue. Many NCI Cancer Centers also have "branch campuses" and mobile treatment infusion options. Be sure and check that out.
Reviewer 2 Report
Comments and Suggestions for Authors
Dear Authors,
thank you for the possibility to read the paper.
I have read the paper, and I will try in the following to provide some comments to improve your research.
Good luck for the publication!

Round 2
Reviewer 1 Report
Comments and Suggestions for Authors
The limitations of this study are clearly identified and are based on the fact that this is a single state study. However, this is apparent to the reader and those reviewing it for replication for future studies. Perhaps the quality of the National Cancer Registry SEER data will improve to allow combined multiple state analyses in the future.
Reviewer 2 Report
Comments and Suggestions for Authors
Dear Authors,
I appreciate a lot the work that you have done to address all the comments.
I think that the integrations allow to have a more complete understanding of your research.